# Rethinking Sesquiterpenoids: A Widespread Hormone in Animals

**DOI:** 10.3390/ijms23115998

**Published:** 2022-05-26

**Authors:** Wai Lok So, Zhenpeng Kai, Zhe Qu, William G. Bendena, Jerome H. L. Hui

**Affiliations:** 1School of Life Sciences, State Key Laboratory of Agrobiotechnology, The Chinese University of Hong Kong, Hong Kong, China; henrysowailok@yahoo.com.hk (W.L.S.); quzheouc@gmail.com (Z.Q.); 2Simon F.S. Li Marine Science Laboratory, School of Life Sciences, The Chinese University of Hong Kong, Hong Kong, China; 3School of Chemical and Environmental Engineering, Shanghai Institute of Technology, Shanghai 201418, China; kaizp@sit.edu.cn; 4Department of Biology, Queen’s University, Kingston, ON K7L 3N6, Canada

**Keywords:** evolution, farnesoic acid, methyl farnesoate, juvenile hormone, insect, metazoan, cnidarian, protostome, deuterostome

## Abstract

The sesquiterpenoid hormone juvenile hormone (JH) controls development, reproduction, and metamorphosis in insects, and has long been thought to be confined to the Insecta. While it remains true that juvenile hormone is specifically synthesized in insects, other types or forms of sesquiterpenoids have also been discovered in distantly related animals, such as the jellyfish. Here, we combine the latest literature and annotate the sesquiterpenoid biosynthetic pathway genes in different animal genomes. We hypothesize that the sesquiterpenoid hormonal system is an ancestral system established in an animal ancestor and remains widespread in many animals. Different animal lineages have adapted different enzymatic routes from a common pathway, with cnidarians producing farnesoic acid (FA); non-insect protostomes and non-vertebrate deuterostomes such as cephalochordate and echinoderm synthesizing FA and methyl farnesoate (MF); and insects producing FA, MF, and JH. Our hypothesis revolutionizes the current view on the sesquiterpenoids in the metazoans, and forms a foundation for a re-investigation of the roles of this important and yet neglected type of hormone in different animals.

## 1. Introduction

### 1.1. Sesquiterpenoids in Insects

Hormones are important regulators of animal development and play important roles in both physiology and body plan diversification during animal evolution [1,2]. Sesquiterpenes are a class of terpenes that consist of three isoprene units, and a sesquiterpenoid hormone named juvenile hormone (JH) is precisely controlled for their titers, together with ecdysteroid, in the processes of complete and incomplete metamorphosis of insects (i.e., holometabolous and hemimetabolous) [1,3,4]. In brief, JH binds to intracellular receptor, Methoprene-tolerant (Met) and triggers the expression of downstream genes, including *Kruppel homolog 1* (*Kr-h1*), that regulate development and metamorphosis in insects [5,6,7] (Figure 1).

Sesquiterpenoids are produced through the conserved mevalonate (MVA) pathway in animals [5,6,7,8]. The first step of sesquiterpenoid biosynthesis occurs in the cytoplasm, under the action of thiolase, two acetyl-CoA molecules combined to form a single acetoacetyl-CoA. Acetoacetyl-CoA then condenses with another acetyl-CoA molecule to form β-hydroxy-β-methylglutaryl-CoA (HMG-CoA), which is catalyzed by HMG-CoA synthase. The second phase begins with the reduction of HMG-CoA to form mevalonate, under the action of HMG-CoA reductase, which is the rate-controlling enzyme. Subsequently, two rounds of phosphorylation occur by mevalonate kinases to form mevalonate diphosphate. Decarboxylation and dehydration reactions then sequentially occur in the conversion of mevalonate diphosphate into isopentenyl diphosphate (IPP). With the addition of geranyl diphosphate, IPP is converted to farnesyl diphosphate (FPP) under the action of farnesyl diphosphate synthase [5,6,7,8]. 

Insects and other arthropods, due to the absence of squalene synthase, do not synthesize cholesterol [9,10]. Alternatively, they possess farnesyl diphosphate phosphatase (FPPP) and farnesol dehydrogenase (FOHSDR), which allow the conversion of farnesyl diphosphate into farnesol (FOH) and farnesal (FO), respectively [11,12,13,14,15]. FO is then oxidized by aldehyde dehydrogenase III, resulting in the formation of farnesoic acid (FA) [16]. FA could be further converted to methyl farnesoate (MF) by the juvenile hormone acid *O*-methyltransferase (JHAMT). Different lineages of insects utilize different end products in this sesquiterpenoid pathway [7]. In certain insects, for instance, the cockroach, grasshopper, and *Drosophila*, further conversion of MF into JH III is possible, under the reaction of an epoxidase, CYP15 [17,18,19]. In the hemipterans, JHSB_3_ is produced by a methyl farnesoate epoxidase (EPOX) from MF [20]. In other insects such as the lepidopterans, FA is first converted into JH III acid with CYP15 and subsequently converted into JH with JHAMT [5,21,22,23]. It has also been demonstrated that FA could be epoxidized and methylated into JHB_3_ in *D. melanogaster* and JHAMT is also suggested to be involved in the methylation of JHB_3_ acid to JHB_3_ [24,25].

Apart from the conventional JH-specific synthesis pathway, another side branch called the isoprenylation pathway was also discovered to produce farnesal [26]. Thus, this pathway could also potentially provide the basis of the synthesis of higher-order sesquiterpenoids. This pathway also utilizes FPP as an initial substrate and through the steps of enzymatic reactions using protein farnesyl transferase (PFT), endopeptidase (ste24), protein-S-isoprenylcysteine O-methyltransferase (ICMT), and prenylcysteine oxidase (PCYOX), farnesal would be produced as a by-product [26].

The biosynthesis of JH has long known to be specific to insects since its discovery in the 1930s by Vincent Wigglesworth [1,6,23,27]. Different insects are now known to have different sesquiterpenoid biosynthetic pathways and modified products [7]. Juvenile hormone III (JH III) has been detected in all insects apart from the hemipterans, which mainly produce JH III skipped bisepoxide (JHSB_3_) [28]. Some dipterans, including *Drosophila*, have also been shown to produce JH bisepoxide (JHB_3_) in addition to JH III [29,30]. Furthermore, apart from JH III, lepidopterans exclusively produce JH 0, JH I, 4-methyl-JH I, and JH II [21,31]. Methyl farnesoate (MF), the precursor of JH III, has also been found in cockroach *Diploptera punctata* nymphs and fly *Drosophila melanogaster* [32,33]. In general, sesquiterpenoids regulate metamorphosis, reproduction, sexual dimorphism, eusociality, and the defense mechanism in insects [7].

### 1.2. Sesquiterpenoids in Non-Insect Arthropods

Despite the lack of juvenile hormone in other arthropods, including crustaceans (crab, crayfish, lobster, shrimp), chelicerates (spider, mite, tick), and myriapods (centipedes and millipedes), other types of functional sesquiterpenoids are present [34,35,36,37,38,39,40,41,42,43,44,45,46]. The existence of sesquiterpenoid MF in crustaceans was first identified in the late 1980s by gas chromatography-mass spectrometry (GC-MS) in the crab *Libinia emarginata* [34] and was since then identified in different groups of crustaceans [35]. MF, in general, has been regarded as the “crustacean JH”, and plays important roles in molting, reproduction, metamorphosis, and respondence to environmental stresses [1,6]. 

Sesquiterpenoids in non-insect/crustacean arthropods such as the chelicerates, on the other hand, are relatively understudied. Enzymes of the mevalonate (MVA) pathway and the downstream pathway leading to the production of MF were identified in the spider mite *Tetranychus urticae*. The presence of MF in *T. urticae* was subsequently confirmed by GC-MS and is likely the end-product of the pathway as the epoxidase to convert MF further was missing in the genome [36]. Through the use of JH analogs and mimics, sesquiterpenoids have been implicated in the development and reproduction of mites, ticks, and spiders [1,37,38,39,40,41,42,43]. In further support of the MVA pathway, and likely sesquiterpenoids being used to regulate reproduction, was the presence and expression pattern of the allatoregulatory neuropeptides allatostatin and allatotropin in the female spider *Parasteadtoda tepidariorum* [44].

In 2014, with the sequencing of the first myriapod genome, the centipede *Strigamia maritima*, genes involved in the biosynthetic pathway leading to sesquiterpenoid production were identified [45]. Recently, a myriapod genome sequencing study (including *Lithobius niger*, *Rhysida immarginata*, *Thereuonema tuberculata*, *Anaulaciulus tonginus*, *Glomeris maerens*, *Niponia nodulosa*) also applied a similar strategy and sesquiterpenoid pathway genes were also found in the genomes. *JHAMT* was found to be absent in the investigated millipede genomes, suggesting FA as the final product of the pathway [46].

### 1.3. Sesquiterpenoids in Non-Arthropods

In the 1990s, exogenous applications of MF and JH induced the settlement and metamorphosis of polychaete *Capitella* [47,48]. In 2016, MF was first detected in head extracts of polychaete *Platynereis dumerilii*. MF application suppressed the expression of the yolk protein gene *Vitellogenin (Vg)* and arrested reproductive maturation [49]. Later, in another polychaete *Syllis magdolena*, MF, along with dopamine and serotonin, was found to trigger the formation of a posterior gamete containing a structure known as a stolon, which upon maturation releases gametes [50]. Recent genome analyses have also revealed the presence of biosynthetic pathway genes in gastropods and bivalves, and in vivo MF treatment could alter the gene expression of sesquiterpenoid pathway genes in the snail *Biomphalaria straminea* [51].

In a recent analysis of jellyfish genomes, the existence of genes involved in the biosynthetic pathway of sesquiterpenoids was also unexpectedly discovered [52]. This study suggested that the sesquiterpenoid hormone system is ancestrally established in the animals. In this article, combining the existing literature and carrying out additional experiments such as hormonal measurement and systematically mining the sesquiterpenoid pathway genes in genomes at key phylogenetic positions, we asked the question: How widespread is the sesquiterpenoid system in different groups of animals?

## 2. Materials and Methods

### 2.1. In Silico Sesquiterpenoid System Gene Analyses and Phylogenetic Analyses

Gene family sequences were first retrieved from the NCBI public database. Each gene was blasted to the published animal genomes, including *Amphimedon queenslandica* (sponge), *Hormiphora californensis* (ctenophore), *Trichoplax adhaerens* (placozoan), *Nematostella vectensis* (seas anemone), *Acropora digitifera* (coral), *Sanderia malayensis* (jellyfish), *Rhopilema esculentum* (jellyfish), *Hydra vulgaris* (hydra), *Helobdella robusta* (leech), *Caenorhabditis elegans* (nematode), *Branchiostoma lanceolatum* (amphioxus), *Branchiostoma floridae* (amphioxus), *Branchiostoma belcheri* (amphioxus), *Ciona intestinalis* (tunicate), and *Strongylocentrotus purpuratus* (sea urchin). The identity of each retrieved gene was subsequently submitted to the NCBI online nr database for reciprocal BLAST using the BLASTP and BLASTX algorithm. The conserved functional domain of the protein was further examined using the NCBI CD search.

For phylogenetic analyses, sequences were first retrieved from the respective animal genomes and aligned to other known animal reference sequences, which were retrieved from NCBI, using ClustalW in MEGA 7.0 software [53]. The best model was predicted using ModelFinder [54] of iqtree v2.2.0-beta [55], and the maximum likelihood (ML) trees and bootstrap trees were built using iqtree with the parameters “--seqtype AA --runs 1 -T AUTO -B 1000 -bnni --alrt 1000”. 

### 2.2. Sesquiterpenoid Hormone Measurement

Freeze-dried jellyfish samples were placed in stainless steel grinding jars with balls of Restch MM400. The grinding jars were chilled in liquid nitrogen for 30 min, and then homogenized at a frequency of 20 Hz for 0.5 min [56]. The homogenate was immediately transferred to a 10 mL glass centrifuge tube containing 1 mL acetonitrile, 1 mL 0.9% (*w*/*v*) sodium chloride solution, and 10 ng JH III-D3 as an internal standard; ultrasonicated for 1 min; and then vortexed and extracted twice with 2 mL hexane. The hexane phase (upper layer) was removed and transferred to a new glass vial, and then dried under nitrogen flow. The residue was dissolved in 1 mL acetonitrile. The measurement of FA was determined using the liquid chromatography method coupled with electrospray tandem mass spectrometry (LC-MS/MS) reported by Ramirez et al. [57].

## 3. Results and Discussion

### 3.1. Farnesoic Acid and Methyl Farnesoate in Non-Insect Protostomes

To systematically identify sesquiterpenoid pathway genes in genomes at key phylogenetic positions, we need to first understand the current animal phylogeny. Cnidaria includes jellyfish, sea anemones, and corals and is the closest relative of bilaterally symmetrical animals, Bilateria. Bilateria comprises three major clades: Deuterostomia (hemichordates, echinoderms, and chordates); Protostomia, which can be further separated into Ecdysozoa (arthropods, nematodes, tardigrades); and Lophotrochozoa (annelids and molluscs). In all the animal genomes searched in this study, all the genes involved in the MVA pathway and isoprenylation pathway could be identified (Figure 2A; Appendix A), suggesting the conservation of these pathways across metazoans.

In chelicerates and crustaceans, the majority of the sesquiterpenoid biosynthetic pathway genes could be identified except the epoxidase genes leading to the synthesis of juvenile hormone [23,58,59] (Figure 2A). MF, the final product, has been demonstrated to be the functional hormone that regulates development, reproduction, and sex determination in crustaceans [60,61,62,63,64,65,66,67,68] (Figure 3). In centipede *Strigamia maritima*, genome sequencing identified the MVA pathway genes and the downstream pathway leading to MF as was found in chelicerates and crustaceans [23,45]. Recently, genes present in the sesquiterpenoid biosynthetic pathway could also be identified in more centipede and millipede genomes [46] (Figure 2A). 

In the other lineage of protostomes, the Lophotrochozoa, a genomic search conducted on the mollusk genomes showed that the sesquiterpenoid biosynthesis genes are present in gastropods and bivalves [51]. In addition, MF could be detected in the annelid and involved in metamorphosis, reproduction, and stolon formation [35,49,50] (Figure 3). These studies suggest that a functional sesquiterpenoid hormonal system with conserved roles in the control of development and reproduction exists in protostomes (Figure 4).

### 3.2. Sesquiterpenoids in Cephalochordate, Urochordate, and Echinoderm

In vertebrates, FPP is converted to squalene by squalene synthase, and subsequently into cholesterol [1,69]. To ask the question whether sesquiterpenoids exist in non-vertebrate deuterostomes, we searched for sesquiterpenoid biosynthetic pathway genes in the genomes of amphioxus *Branchiostoma floridae* and *Branchiostoma belcheri*, sea urchin *Strongylocentrotus purpuratus*, and tunicate *Ciona intestinalis* as representatives of cephalochordate, echinoderm, and urochordate, respectively.

To our surprise, aldehyde dehydrogenase (ALDHIII) could be identified in the genomes of *B. floridae*, *B. belcheri, S. purpuratus*, and *C. intestinalis*, suggesting sesquiterpenoid farnesoic acid (FA) could be produced in these animals (Figure 2A; Appendix A). In addition, the juvenile hormone *O*-methyltransferase (JHAMT) could also be identified in the genomes of amphioxus and echinoderms, suggesting that sesquiterpenoids MF may be well produced (Figure 2B and Figure 4; Appendix A). Although FA and MF are not produced in vertebrates, farnesol-like sesquiterpenoids are produced and have been suggested as potential “JH” in vertebrates [70]. These findings suggest that sesquiterpenoid is widely produced in bilaterians. 

### 3.3. Farnesoic Acid in Cnidarians

Previous study revealed sesquiterpenoid biosynthetic pathway genes in jellyfish and suggested that the pathway would terminate with the production of FA [52] (Figure 2A and Figure 4). Here, we also measured sesquiterpenoids using liquid chromatography coupled with electrospray tandem mass spectrometry (LC-MS/MS), and confirmed that FA was produced at titers of 55.02 ± 9.21 ng/g FA in the medusa of lion mane jellyfish *Cyanea capillata*, and 1.04 ± 0.48 ng/g FA in the medusa of edible jellyfish *Rhopilema esculentum*. 

We further searched for the sesquiterpenoid biosynthetic pathway genes in other cnidarian genomes, including the jellyfish *Sanderia malayensis* and *Rhopilema esculentum,* the sea anemone *Nematostella vectensis*, corals *Acropora digitifera*, *Stylophora pistillata* and *Pocillopora damicornis*, and hydroid *Hydra vulgaris*, and found the presence of MVA, isoprenylation, and ALDHIII genes in all these cnidarian genomes (Figure 2A; Appendix A). This finding suggests that sesquiterpenoids are widely produced in cnidarians, and further consolidates that the cnidarian-bilaterian ancestor had an established sesquiterpenoid system.

### 3.4. Sesquiterpenoids in Early Branched Metazoans?

To explore whether the sesquiterpenoid system could be dated back to the metazoan ancestor, we also investigated the genomes of poriferan *Amphimedon queenslandica* and ctenophore *Hormiphora californensis*. Both *A. queenslandica* and *H. californensis* possess the complete set of genes involved in the terpenoid backbone synthesis and isoprenylation pathways, suggesting the potential presence of farnesal in these animals (Figure 2A; Appendix A). In addition, the presence of ALDHIII could also be identified in these animals, suggesting the likelihood of the conversion of farnesal into farnesoic acid (Figure 2A). Future hormone measurement in poriferans and ctenophores will be needed to test this hypothesis. 

## 4. Conclusions

Here, we hypothesize that the sesquiterpenoid hormonal system is an ancestral system established in an animal ancestor and remains widespread in many animals. Different animal lineages have adapted different enzymatic routes from a common pathway that produces different sesquiterpenoid hormones. Much work needs to be conducted to test the function of these sesquiterpenoid compounds and discover their various roles in different animal lineages.

## Figures and Tables

**Figure 1 ijms-23-05998-f001:**
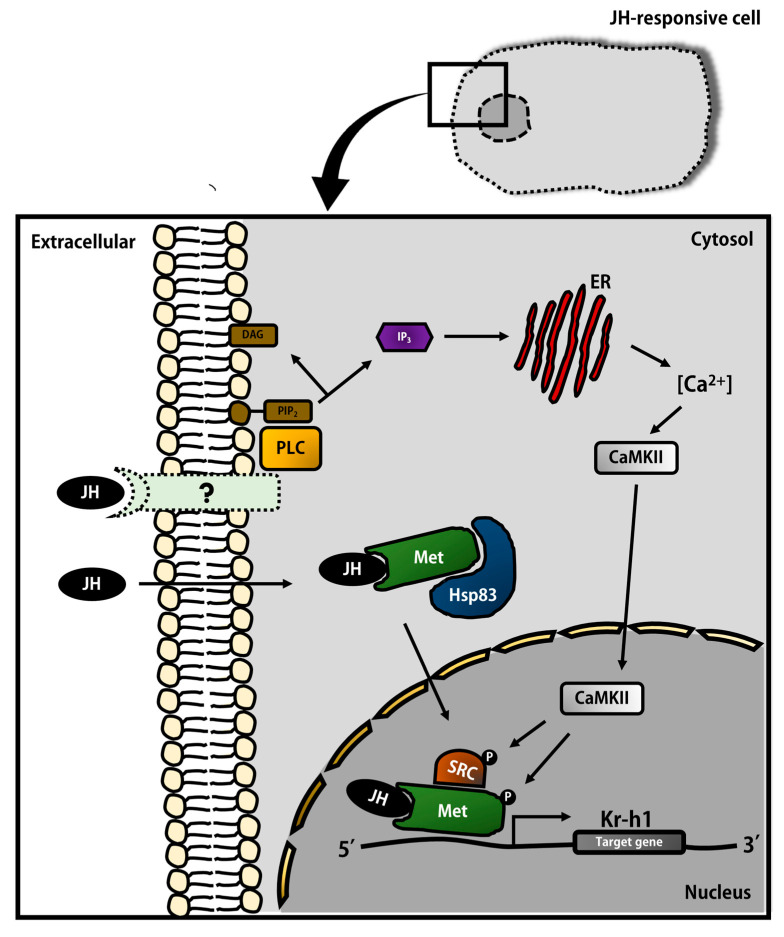
JH signaling pathway.Schematic diagram showing the cellular response upon JH stimulation, summarized from the previous literature. JH binds to its intracellular receptor Methoprene-tolerant (Met) and the complex is transported into the nucleus, mediated by heat shock protein 83 (Hsp83). Steroid receptor coactivator (SRC) then forms a heterodimer with the JH-Met to form an active complex to regulate the transcription of target genes. A putative transmembrane receptor (labeled in pale green with a dotted line) is hypothesized from previous studies, which demonstrated an activated intracellular RTK-signaling pathway (phospholipase C (PLC), phosphatidylinositol biphosphatein (PIP_2_), diacylglycerol (DAG), inositol trisphosphate (IP_3_), and Ca^2+^/calmodulin-dependent protein kinase II (CaMKII)) in JH-stimulating cells.

**Figure 2 ijms-23-05998-f002:**
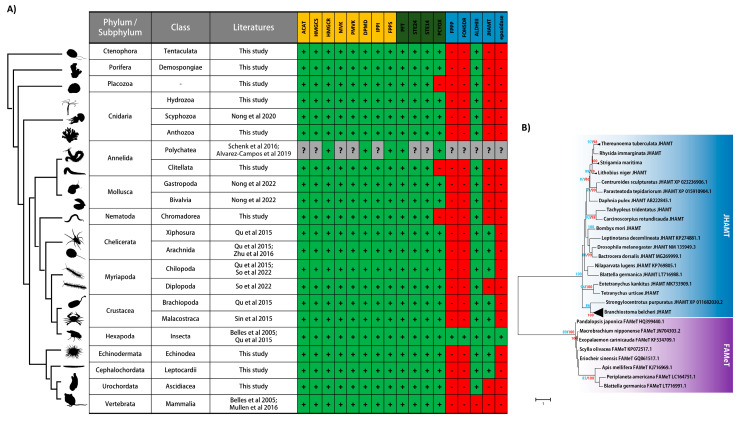
A summary of the literature reporting the gene cassette required in sesquiterpenoid biosynthesis and the *JHAMT* gene tree. (**A**) A table showing the presence of MVA, isoprenylation and JH-specific pathway in different animals, summarized from the previous studies [21,23,46,49,50,51,52,58]. The box labeled in green with “+” indicates the presence of genes; The box labeled in red with “-” indicates the absence of genes. The box labeled in grey with “?” indicates the uncertainty. The genes present in the mevalonate pathway, the isoprenylation pathway, and the downstream juvenile hormone pathway are highlighted in orange, green, and blue, respectively. *ACAT*, Acetyl-CoA Acetyltransferase; *HMGCS*, hydroxymethylglutaryl-CoA synthase; *HMGCR*, 3-hydroxy-3-methylglutaryl-CoA reductase; *MVK*, mevalonate kinase; *PMVK*, phosphomevalonate kinase; *DPMD*, diphosphomevalonate decarboxylase; *FPPS*, farnesyl diphosphate synthase; *FPPP*, farnesyl diphosphate phosphatase; *FOHSDR*, farnesol dehydrogenase (short-chain dehydrogenase); *ALDHIII*, aldehyde dehydrogenase 3; *PFT*, protein farnesyl transferase; *STE24*, endopeptidase; *ICMT*, protein-S-isoprenylcysteine O-methyltransferase; *PCYOX*, prenylcysteine oxidase. (**B**) Phylogenetic gene tree of *JHAMT*s identified in animals. The tree topology shown is constructed by the maximum likelihood (ML) algorithm. The phylogenetic trees were constructed with the LG + G + I model using the maximum likelihood (ML) and neighbor-joining (NJ) methods, rooted with arthropod farnesoic methyltransferase (*FAMeT*) in MEGA 7.0, with 1000 replicates. Only bootstrap values larger than 80% are indicated for clarity (blue from ML and red from NJ).

**Figure 3 ijms-23-05998-f003:**
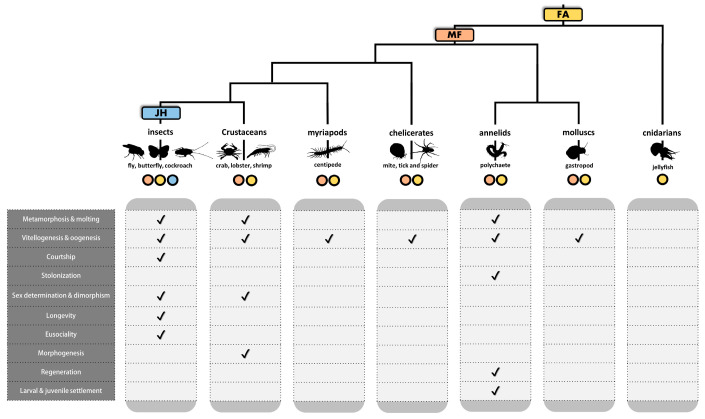
The form of sesquiterpenoids present across the animal phylogeny and their biological effects documented in previous studies.

**Figure 4 ijms-23-05998-f004:**
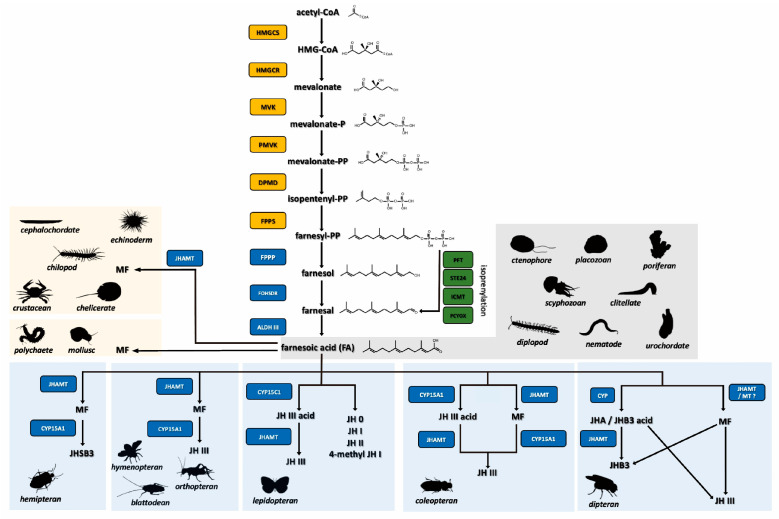
A summary of the sesquiterpenoid biosynthetic pathway and the putative functioning final products in different researched animals up to date. The enzymes in the mevalonate pathway, the isoprenylation pathway, and the downstream JH-specific pathway are highlighted in orange, green, and blue, respectively.

## Data Availability

This study does not report any data.

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
