# Peer review of "Rethinking Sesquiterpenoids: A Widespread Hormone in Animals"

_ijms, 2022, doi:10.3390/ijms23115998_

Round 1
Reviewer 1 Report
Sesquiterpenoids such as juvenile hormones and methyl farnesoate play important roles as hormones controlling development, metamorphosis, reproduction, and others in insects and crustaceans. It has been believed that sesquiterpenoid hormones exist only in these limited groups of Arthropoda. Recently, however, finding sesquiterpenoid biosynthetic genes in genomes and evidences of existence of sesquiterpenoids in other animals are changing our understanding about sesquiterpenoids. In this manuscript, authors spread animal groups in which sesquiterpenoids might function as hormones over Eumetazoa and discuss their evolution using the information from the literatures and the present study. Although this manuscript updates our knowledge and provides us with new insights on sesquiterpenoids in animals, this reviewer feels several concerns as follows:
- Lack of farnesyl diphosphate phosphatase and farnesol dehydrogenase
Authors’ opinions about existence of sesquiterpenoids are based on the existence of biosynthetic genes in the genomes and the detection of sesquiterpenoids from the particular animals. Figure 1 shows, however, all animal groups except for insects lack the farnesyl diphosphate phosphatase and farnesol dehydrogenase, which catalyze the first two steps of the sesquiterpenoid biosynthetic pathway. It is necessary to explain why animals can produce sesquiterpenoids without these enzymes.
- Reliability of evidences
Some evidences for the existence of biosynthetic genes in the genomes and the detection of sesquiterpenoids are not clearly shown. For example, according to Figure 1 the biosynthetic genes of sesquiterpenoids in cephalochordate, urochordate, and echinoderm seem to be found in the current study. Although the phylogenetic tree of JHAMT is shown in Figure 2, the information of other genes is not provided. In the section 2.3, authors describe about detection of FA in cnidarians. If this is not from a literature but a novel data, the experimental procedure and the result should be provided in detail.
Figure 3 shows MF in myriapods. What is the evidence for this? It is said that FA might be the final product in millipede in line 97. In line 85, it is described that the presence of MF in the spider mite was confirmed by GC-MS with the citation #30; but the #30 does not seem to describe the detection of MF.
Other minor concerns are listed below:
- Line 50
According to my understanding, farnesal is not de-methylated but oxidized when it is converted to farnesoic acid.
- Line 83
“Enzymes of the mevalonate (MVA) pathway leading to the production of MF” is not appropriate. The mevalonate pathway generally indicates the steps from acetyl-CoA to farnesyl pyrophosphate. Similar expression is also seen in line 138.
- Line 97
It is not generally appropriate to cite a paper that is under review. It should be indicated as “submitted” or “personal communication” without journal name.
- Line 126
The term of “isoprenylation pathway” might not be well known. It should be explained.
- Figure 1
The abbreviations for enzymes/genes are not described, although they are mentioned in Figure 4. They should be explained in Figure 1 or in the abbreviation section. The color codes for enzymes/genes should be explained, too.
- Line 139
“Complete set” is not correct expression because farnesyl diphosphate phosphatase and farnesol dehydrogenase are lacking.
- Figure 4
The dotted arrow with the enzymes of “isoprenylation pathway” should be explained.
- Line 188-191
This reviewer does not understand clearly this description. What kind of event(s) are assumed? And did deuterostomes evolve from prostostomes?
Reviewer 2 Report
Although the main text is scant, I think that it provides a good overview of Sesquiterpenoids in many classes.
I suggest carefully reading the text as many English mistakes occurred.
Please delete these keywords as they are already in the title: Hormone; sesquiterpenoid
Please add references to these sentences:
-“Hormones are important regulators of animal development and play important roles in both the physiology and body plan diversification during animal evolution.”
-“The first step of sesquiterpenoid biosynthesis …. under the action of farnesyl diphosphate synthase.”
-“In certain insects, for instance the cockroach, grasshop-54 per and Drosophila, they are able to further convert MF into JH III, under the reaction of 55 an epoxidase, CYP15.”
-“Different insects are now known to have 63 different sesquiterpenoid biosynthetic pathways and modified products”
-“Despite the lack of juvenile hormone in other arthropods, including crustaceans (crab, crayfish, lobster, shrimp), chelicerates (spider, mite, tick) and myriapods (centipedes and milliepdes), there are presence of other types of functional sesquiterpenoids.”
-“Recently, a myriapod genome sequencing study also applied a similar strategy and sesquiterpenoid pathway genes were also found in the genomes.” Which myriapod?
“In vertebrates, FPP is converted to squalene by squalene synthase, and subsequently into cholesterol (Figure 4).”
- “Sesquiterpenes are a class of terpenes that consist of three isoprene units, and a sesquiterpenoid hormone named as juvenile hormone (JH) is precisely controlled for their titers in the processes of complete and incomplete metamorphosis of insects (i.e. holometabolans and hemimetabolans)”. Moreover, I suggest to write “juvenile hormone (JH) is precisely controlled for their titers, together with ecdysone, in the processes of complete and incomplete metamorphosis of insects” (https://doi.org/10.1016/j.jinsphys.2018.02.008; https://doi.org/10.1016/j.cub.2019.10.009), and please substitute “holometabolans and hemimetabolans” with “holometabolous and hemimetabolous”.
In figure 1 I suggest writing the references with numbers, in this way it is easier for the reader to find the right reference.
In the conclusion section the authors should highlight their hypothesis and their “findings”.
Round 2
Reviewer 1 Report
The current manuscript has been revised very well. This reviewer is pleased to agree to publish this study. Several minor points to be revised were found as listed below:
1. affiliations
Although three affiliations are shown, the third one is related to no one.
2. Line 15
“mine” might be “ours”, and “for” could be necessary after the “ours”.
3. Line 20
“synthesise” might be “synthesizing”.
4. Line 36
Figure 2 is mentioned earlier than Figure 1. This is not appropriate. The Figure 2 also precedes Figure 1.
5. Line 101
“milliepdes” must be “millipedes”.
6. Line 222
Cholesterol biosynthesis in vertebrates is described with Figure 4. But, Figure 4 does not show Cholesterol biosynthesis in vertebrates.
7. Figure 4
Although FPPP, FOHSDR, and ALDHIII are colored orange, they are enzymes in “the downstream JH-specific pathway”. Therefore, they should be colored blue.
8. Supplementary Figure 1-12
These supplementary figures were included in “Non-published Material”. They should be published as supplementary figures.
